# A Systematic Review of Tear Vascular Endothelial Growth Factor and External Eye Diseases

**DOI:** 10.3390/ijms25031369

**Published:** 2024-01-23

**Authors:** Jaclyn Chan, Gavril Lim, Ryan Lee, Louis Tong

**Affiliations:** 1Edinburgh Surgery Online, The University of Edinburgh, Room G10/G11, Simon Laurie House, 196 Canongate, Edinburgh EH8 8AQ, UK; 2Training and Education Department, Singapore National Eye Centre, Singapore 168751, Singapore; 3Lee Kong Chian School of Medicine, Nanyang Technological University, Singapore 639798, Singapore; 4Corneal and External Diseases Department, Singapore National Eye Centre, 11 Third Hospital Avenue, SNEC, Building, Singapore 168751, Singapore; 5Ocular Surface Group, Singapore Eye Research Institute, Singapore 169856, Singapore; 6Eye Academic Clinical Program, Duke-NUS Medical School, Singapore 169857, Singapore

**Keywords:** human, review, ocular surface disease, tear disorders, dry eye, VEGF, atopy

## Abstract

We aim to summarize the current evidence of Vascular endothelial growth factors (VEGF)s in external eye diseases and determine whether serum and plasma VEGF levels are associated with tear and ocular surface tissues. A systematic search of PUBMED and EMBASE was conducted using PRISMA guidelines between October 2022 and November 2023, with no restriction on language or publication date. Search terms included relevant MESH terms. These studies were evaluated for quality, and an assessment of the risk of bias was also carried out. Extracted data were then visually represented through relevant tables or figures. The initial literature search yielded 777 studies from PUBMED, 944 studies from EMBASE, and 10 studies from manual searches. Fourteen eligible studies were identified from 289 articles published from 2000 to 2023 in the English language or with English translations, including rabbit models, murine models, and human-derived samples. Most studies were retrospective in nature and case–control studies. Various common external eye diseases, such as dry eye disease (DED) and allergic eye disease were investigated. Despite limitations and small sample sizes, researchers have found elevated tissue levels of the VEGF in the vascularized cornea, especially in animal models, but there is no evidence of clear changes in the tear concentrations of VEGF in DED and allergic eye disease. Tear VEGF is associated with corneal vascularization. Anti-VEGF therapies may have the potential to manage such conditions.

## 1. Introduction

The normal ocular surface system consists of the eyelids, eyelashes, cornea, conjunctiva, lacrimal glands, meibomian glands, and nasolacrimal duct. These structures are integrated by the continuous epithelia and homeostasis is regulated by various nervous, endocrine, immune, and vascular systems [1,2]. Dysfunction or dysregulation in any of these structures and systems leads to a spectrum of eye diseases, collectively described as ocular surface disease (OSD). OSD, including dry eye disease (DED), thyroid eye disease (TED), Steven–Johnson syndrome (SJS), and mucous membrane pemphigoid, are associated with a significant impact on vision-related daily activities and reduced quality of life [3,4,5]. Autoimmunity and inflammation play a significant role in the pathogenesis of many OSDs, and they are responsible for a significant amount of eye-related morbidity in the world, including blindness. Thus, identifying the key molecules/pathways involved in disease pathogenesis may potentially provide new targets for treatment [6]. Vascular endothelial growth factors (VEGFs) are important regulators of angiogenesis [7,8,9,10,11] and have been implicated in inflammation. A schematic illustration of VEGFs and their receptor system is shown in Figure 1. Figure 2 illustrates selected ocular surface diseases and VEGF treatments.

Pterygia are benign surface ocular lesions that start growing from the limbal epithelium and slowly invade the cornea centripetally followed by conjunctival epithelium [12]. It is believed that nitric oxide expressed is increased via UV light, which results in the elevation of VEGF levels in human endothelial cells, promoting pterygia’s angiogenic properties [13]. DED is known to be characterized by several key cellular events, including conjunctival goblet cell loss, epithelial cell abnormality, and tear secretion deficiency [14]. Essentially, DED is the outcome of many factors that result from the inflammation of the cornea and conjunctiva, as well as decreased tear production or excessive evaporation [15]. Eye injuries may result in permanent corneal damage via intraocular inflammation, leading to opacification and the loss of visual acuity. It has been suggested that the presence of herpes simplex keratitis (HSK) leads to the inhibition of sVascular Endothelial Growth Factor Receptor-1 (sVEGFR-1) synthesis at a higher rate than the VEGF, resulting in a ratio imbalance between sVEGFR-1 and VEGF; hence, the release of VEGF is accelerated to consequently cause angiogenesis [16]. Allergic eye disease comprises antibody-mediated disorders and/or T-lymphocyte-mediated disorders where the adaptive immune response has been mounted against allergens in the eye [17]. The pathogenesis of OSD in TED patients is multifactorial in nature; structural changes resulting in increased ocular surface exposure and altered blink dynamics have been linked to evaporative dry eye disease. Concurrently, tear-film changes and the overexpression of pro-inflammatory cytokines have also been implicated in the inflammatory cascade leading to OSD [18]. A wide variety of ocular insults can lead to abnormal new vessels invading the corneal stroma because of disruption in the balance of angiogenic and antiangiogenic factors, leading to corneal neovascularization [19,20]. The VEGF has been reported to be a promoter of ocular surface neovascularization [21,22]. Furthermore, factors that drive DED also increase the expression of factors that promote lymphangiogenesis [23]. There have been reviews of inflammation and dry eye disease [10] and suggestions of inflammation as a measure of clinical disease in dry eye [24,25]. A recent systemic review has also explored corneal lymphangiogenesis as a target in DED [26].

One of the most significant therapeutic advances in ophthalmology in the last two decades is the use of commercially available anti-VEGF agents. There is also an ongoing race in the industry to develop tear-based point-of-care devices that can assay biomarkers such as VEGF to be used in clinics in a noninvasive or minimally invasive way, to guide clinical decisions. It is therefore relevant and important to examine the role of VEGF in ocular surface disease. Based on our literature search, there has been no systematic review that has comprehensively summarized the existing literature on the role of VEGF in OSD and its potential as a target for treatment.

By summarizing the current evidence of VEGFs in OSDs and determining whether the serum and plasma VEGF levels are associated with tear and ocular surface tissues, this review aims to explore whether VEGF is a useful biomarker for patients with external eye diseases and whether anti-VEGF intervention has a role in treating external eye diseases.

## 2. Methods

### 2.1. Search Strategy

This search was conducted in line with the Preferred Reporting Items for Systematic Reviews and Meta-Analyses (PRISMA) statement. Searches were conducted using PUBMED and EMBASE between October 2022 and November 2023. Search terms included “vascular endothelial growth factor” [MeSH Terms] OR “vascular endothelial growth factor” [Text Word]; “cornea neovascularization” [MeSH Terms]; “dry eye syndrome” [MeSH Terms]; “dry eye syndrome”, “dry eye disease”, “Keratoconjunctivitis Sicca”, “Xerophthalmia”; “corneal lymphangiogenesis” [MeSH Terms]; “external eye disease” [MeSH Terms]. Several permutations were introduced during the systemic search to ensure the full coverage of the literature.

There were no restrictions on language or publication date. However, as no additional translational services were available, foreign language studies were excluded if contacting relevant authors was unsuccessful. Bibliographies of the selected articles and studies were also reviewed. Study authors/organizations did not have to be contacted via email for further clarification. An intended response time of four weeks was taken into consideration if contact was necessary.

Studies identified after the initial search were screened to determine whether they presented data concerning VEGFs in external eye diseases. The title, abstract, and finally full text were screened to assess the discussion of VEGFs and external eye diseases. For inclusion, studies must also have presented original data. Figure 3 illustrates the literature search process.

### 2.2. Criteria

A total of 14 articles, published from 2000 to 2023 in the English language or with English translations, were included. Other articles published prior to 2010 were cited for foundational knowledge on the topic. Animal and human studies, published in English (full text), were included in this review. Studies were included if they fulfilled the following criteria: (1) observational research (e.g., cross-sectional studies, case series) or interventional research (e.g., randomized controlled trials), and (2) investigated serum and/or plasma and/or tear VEGF levels.

The additional inclusion criteria were (1) studies in adult subjects (age ≥ 18 years); (2) the study population consisting of patients with external eye diseases; (3) improvement in external eye disease as an outcome measure and the control group consisting of nonexternal eye disease patients; and (4) studies published in English.

The exclusion criteria were as follows: (1) samples that overlapped with another study, or (2) review articles, case reports, letters to the editor, conference abstracts, or in vitro studies. Studies that met the inclusion criteria were evaluated for quality, and an assessment of the risk of bias was also carried out following the methodology of Cochrane Collaboration or Consolidated Standards of Reporting Trials [27] to ensure validity and reliability.

### 2.3. Ethics and Declarations

The research described in this project was formally approved by the University of Edinburgh. Additional institutional review board approval is not required for systematic review in Singapore. The authors report no commercial or proprietary interest in any product or concept discussed in this review.

### 2.4. Assessment of Quality, Data Extraction, and Risk-of-Bias Assessment

The selected studies were evaluated for quality using the Critical Appraisal Skills Programme (CASP) checklist [28]. An assessment of the risk of bias was carried out following the methodology of Cochrane Collaboration or Consolidated Standards of Reporting Trials [27] to ensure validity and reliability. The risk of bias for each study was also gathered in a table, and an overall assessment of the risk of bias was carried out.

## 3. Results and Discussion

### 3.1. Study Characteristics

The initial literature search yielded 777 studies from PUBMED, 944 studies from EMBASE, and 10 studies from manual searches, which included reviewing the reference lists and/or bibliographies of the selected studies. Duplicates were removed, and articles were screened by reviewing their dates, titles, abstracts, and full texts. In total, 14 eligible studies were identified from the 289 articles published from 2000 to 2023 in the English language or with English translations. An overview of the included studies and their respective findings is presented in Table 1.

Table 2 outlines each individual study’s characteristics. A mix of rabbit models, murine models, and human-derived samples was used in the selected studies. Most studies were retrospective in nature and case–control studies. Two studies were age/gender-matched due to the limited number of samples. The studies were conducted in several countries, ranging from the US to Asia. Various common external eye diseases were investigated, such as dry eye disease and corneal neovascularization.

### 3.2. Overall Trend and General Findings

In the findings related to VEGF in various ocular tissues, the unit of VEGF measurement was uniformly converted to pg/mL for comparison between studies, where possible. Six out of the fourteen studies involved human samples or participants. One study involved cell cultures with herpes simplex viral infection, and the other seven studies were animal-based. The results were further classified into various OSDs.

#### 3.2.1. Studies in Pterygium

Aspiotis et al. retrospectively studied fifty-two surgically excised pterygia and seven normal conjunctivae and demonstrated that pterygium tissues were more vascularized than normal conjunctivae and the overexpression of the VEGF in pterygium endothelial and stromal cells [29]. Similarly, Van Acker et al. [30] demonstrated an increase in VEGF levels in patients’ unaffected eye and recurrent pterygium groups (*p* < 0.05).

#### 3.2.2. Studies in Corneal neovascularization

Corneal vascularization was due to a variety of causes, including previous bacterial infection, aniridia, chemical burns, and idiopathic limbal stem cell deficiency. Philipp et al. confirmed that the VEGF was expressed by epithelial cells, corneal endothelial cells, the vascular endothelial cells of limbal vessels, and some keratocytes [31]. This study also revealed that the amount of VEGF protein in vascularized corneas (109.4 ± 86.8 pg) was significantly higher than that (10.0 ± 4.3 pg) of the normal control corneas of the same size (half-corneal buttons, 7 mm in diameter; *p* < 0.01). Zakaria et al. compared the basal and reflex tear VEGF levels of twelve patients with vascularized cornea in one of their eyes (six males and six females) with that of ten age-matched healthy controls. The tear VEGF levels were found to be significantly higher in the eyes with vascularized corneas than in their healthy eyes and the eyes of healthy controls [32].

#### 3.2.3. Studies in Dry Eye Disease

The study by Bradley et al. is the only study that investigated human serum VEGF levels (68.76 ± 38.70 pg/mL) in DED patients [33]. Byambjav et al. found tear VEGF levels to be similar between groups of participants: healthy controls, DED only, diabetics, and diabetics with DED (*p* > 0.5) [34].

#### 3.2.4. Studies in Corneal Infection

The in vitro and in vivo studies of herpes simplex virus (HSV)-1-infected corneal epithelial cells by Wuest et al. suggest that HSV-1 infection drives the transcriptional upregulation of VEGF-A and that HSV-1-infected cells are the predominant source of VEGF-A during acute HSV-1 infection [35]. The rabbit model used in the study by Mastyugin et al. confirmed the presence of VEGF mRNA in corneal epithelium, and a resultant densitometry analysis proved a 9- to 12-fold increase in VEGF levels secondary to hypoxic injury [36]. Goswami et al. utilized a rabbit model and found a significant increase in the expression of VEGF beginning on day 1 after sulfur mustard (SM) exposure, which remained significantly elevated till day 14 post-exposure (*p* < 0.05) [37]. The same model also highlighted that the maximal increase in the VEGF expression as compared to control was observed on day 14 post-exposure for both the 5 min and 7 min exposure durations (~6.0- and 4.6-fold increases, respectively). Narimatsu et al. utilized a murine model to simulate bacterial keratitis and illustrated that angiogenesis increased significantly on day 7 (infected: 38.52% versus control: 15.44%, *p* < 0.05) and day 14 (infected: 57.57% versus control: 11.34%, *p* < 0.05) post-inoculation in the infected group compared to controls [38]. Another similar murine model by Xue et al. also confirmed that VEGF expression was significantly (*p* < 0.05) elevated after bacterial infection [39].

#### 3.2.5. Studies in Allergic Eye Disease

We did not find any human studies that evaluated VEGF and allergic eye disease. In a murine study, Lee et al. studied corneas in allergic eye disease, which showed a significant upregulation of the lymphangiogenic ligands VEGF-C (*p* < 0.05) and VEGF-D (*p* < 0.01), as well as a nearly seven-fold increase in mRNA levels of VEGFR-3 (*p* < 0.001) [40].

#### 3.2.6. Studies in Corneal Lymphangiogenesis

Lymphangiogenesis in the murine model used by Narimatsu et al. only increased significantly in the infected group on day 14 post-inoculation (infected: 12.00% versus control: 5.17%, *p* < 0.05) [38]. Goyal et al. illustrated an overall increase in VEGF-A, -C, and -D (*p* < 0.03) [41]. Interestingly, the increase in the levels of VEGF-A, VEGF-C, and VEGFR-2 seen in Goyal et al.’s murine model occurred at later time points (day 14), and VEGF-D and VEGFR-3 were increased as early as day 6 of the disease. Mimura et al. simulated corneal chemical injury and demonstrated similar results, where both VEGF-C and its receptor VEGFR-3 were upregulated 3 days after the injury [42].

### 3.3. Risk-of-Bias Assessment

Risk-of-bias assessment was performed using the Risk Of Bias in Nonrandomized Studies of Exposure (ROBINS-E) [43], and the overall results are presented in Figure 4 and Figure 5.

Seven kinds of bias were specifically evaluated (Figure 4), and the studies collectively did not suffer from any one type of bias to a great extent. Nevertheless, the risk-of-bias assessment showed some concerns in 10/14 studies (Figure 5). Only 2 out of the 14 studies showed a high risk of bias.

### 3.4. Discussion

This systematic review has revealed that VEGF levels are correlated with mainly ocular neovascularization from various OSDs, including bacterial keratitis, chemical injury, pterygium, allergic eye disease, and corneal neovascularization. There is insufficient evidence to show that VEGF is linked to the presence of these OSDs themselves, though many studies suffered from small sample sizes and were not designed primarily to assess VEGF levels. In addition, animal studies show interesting results, which implicate VEGF in bacterial infections and allergic eye disease.

#### 3.4.1. General Findings

Overall, human-derived samples ranged from various tissues and tears. Only one study had serum samples. The age of human participants across the relevant studies ranged from 23 to 84 years old, with a good ratio of males and females in most studies. Most studies were retrospective in nature and case–control studies. Two studies were age/gender-matched due to the limited number of samples.

#### 3.4.2. VEGF in Pterygium

Aspiotis et al. [29] demonstrated an overexpression of VEGF in pterygium, in particular endothelial and stromal cells, which is consistent with previous reports [13,20,44] that angiogenesis is a potentially important pathogenic contributor to pterygium formation. The authors also postulate that the overexpression of angiogenic factors and a concurrent decrease in the expression of angiogenesis inhibitors serve as pathways for the pathogenic mechanism of pterygium [29]. Van Acker et al. [30] investigated tear VEGF, IL-6, and IL-8 levels and correlated them with the vascularization and size of pterygia. VEGF levels in pterygia per se were not evaluated. The tear levels of IL-6 were significantly elevated in the primary and recurrent pterygia compared to healthy controls. The tear levels of IL-8 and VEGF showed a more complicated relationship with pterygium status, and the statistically significant findings could have been driven by outliers. It would have been useful to have the analysis repeated excluding the outliers.

#### 3.4.3. VEGF in Dry Eye Disease

The study by Bradley et al. [33] was the only selected study that investigated serum VEGF levels. Although this pilot study did not illustrate any statistically significant differences in the serum levels of VEGF in DED patients versus age-matched controls, it did highlight the need for further investigation with a larger patient sample [33]. Goyal et al. [41] simulated DED and concluded that lymphangiogenic-specific VEGF-D and VEGFR-3 levels increased earliest on day 6, followed by increased VEGF-C, VEGF-A, and VEGFR-2 levels.

In order to enhance our understanding of the relevance of tear VEGF, a larger study was conducted in Singapore, where 102 adult participants (73 female, 29 male) with a mean age of 51 ± 17 years were recruited from the Singapore National Eye Center and underwent a standardized evaluation of the ocular surface, which included an assessment of dry eye symptoms using a validated questionnaire (Standard Patient Evaluation of Eye Dryness Questionnaire (SPEED)) and clinical signs (corneal fluorescein staining, tear break-up times, the Schirmer I test, the tear meniscal height, and conjunctival redness) as well as a standardized assessment of various risks and associated factors of DED. Tear proteins were extracted from the Schirmer strips, and the total protein was quantified. VEGF quantification was performed using the bead-based indirect fluorescent assay (Beadlyte, Bioplex). It was found that female tended to have higher levels of tear VEGF than male, with 25.1 ± 22.5 ng/mL and 22.4 ± 16.1 ng/mL, respectively, but the difference was not statistically significant (*p* = 0.56). Most importantly, the tear VEGF concentration was higher (*p* = 0.039) in participants with a history of atopy (n = 14) than in those without such history (n = 88), with concentrations of 26.9 ± 29.7 (median 25.8, range 4.96–103.3) ng/mL compared to 23.3 ± 18.6 (median 19.3, range 0.57–138.3) ng/mL, respectively. Additionally, the tear VEGF concentration was not associated with age, dry eye symptoms, corneal staining, tear break-up times, the Schirmer I test, the tear meniscal height, and conjunctival redness. We conclude that tear VEGF levels are elevated in participants with a history of atopy. It is likely that such patients have a heightened immune system and are more prone to ocular surface inflammation, which can be triggered by intrinsic or environmental factors.

In a similar study performed in the UK by Byambajav et al. [34], patients were classified into four groups, and it was found that tear fluid IL-6 and IL-8 concentrations correlated with various clinical signs of dry eye in Type 2 diabetes mellitus with DED. The study also concluded that inflammation is an important cause and influences the progression of DED, with the concomitant occurrence of cytokine production [34].

#### 3.4.4. VEGF in Corneal Infection

In vitro and in vivo studies of HSV-1-infected corneal epithelial cells by Wuest et al. [35] reveal that HSV-1 infection drives the transcriptional upregulation of VEGF-A and that HSV-1-infected cells are the predominant source of VEGF-A during acute HSV-1 infection. Additionally, VEGF-A transcription does not require TLR signaling or MAP kinase activation and is directly activated by the HSV-1-encoded immediate early transcription factor, ICP4, resulting in the neovascularization characteristic of ocular HSV-1 disease. Interestingly, corneal lymphangiogenesis via macrophages plays an important role in the resolution of bacterial keratitis, especially in the late stage [38]. A murine model used to simulate bacterial (*Pseudomonas aeruginosa*) keratitis was utilized by Narimatsu et al. [38], illustrating that angiogenesis and lymphangiogenesis were both involved in bacterial keratitis, which is consistent with the reports from various other corneal injury models [13,39,44,45]. Mimura et al. simulated corneal chemical injury using a murine model and demonstrated similar results where both VEGF-C and its receptor VEGFR-3 were upregulated 3 days after the injury [42].

#### 3.4.5. VEGF in Non-Infective Corneal Vascularization

Philipp et al. found that there was higher VEGF expression by corneal epithelial cells, corneal endothelial cells, the vascular endothelial cells of limbal vessels, and some keratocytes in inflamed human corneas from frozen sections as compared to healthy controls [31], suggesting a role of VEGF in inducing and maintaining corneal neovascularization [46]. Human tears analysis by Zakaria et al. highlighted that the concentrations of pro-angiogenic cytokines in human tears were significantly higher than their concentrations in serum, with the highest levels found in the basal tear group [32] and also confirmed VEGF as one of the main mediators in corneal angiogenesis.

#### 3.4.6. VEGF Levels in Animal Models Related to Corneal Vascularization and Lymphangiogenesis

In general, the studies in this review support the idea that VEGF drives angiogenesis and, in turn, inflammation levels, resulting in unwanted side effects, including corneal neovascularization. A murine model by Xue et al. [39] reported significantly higher levels of MIP-2 and VEGF in vascularized corneas compared to normal control corneas, confirming the importance of VEGF as an angiogenic factor in pathological neovascularization. The rabbit model used in the study by Mastyugin et al. demonstrated the presence of VEGF mRNA in the corneal epithelium and a resultant increased expression secondary to hypoxic injury [36], confirming the neovascularization response induced by a prolonged hypoxic state. Goswami et al. utilized a rabbit model and simulated corneal injury using sulfur mustard (SM) exposure, confirming inflammation as a critical factor and that VEGF can be secreted upon vesicant-induced inflammation by macrophages and corneal cells [37].

Goyal et al. [41] concluded that low-grade inflammation associated with DED is an inducer of lymphangiogenesis without accompanying hemangiogenesis. Mimura et al. also illustrated that VEGF-C immunoreactivity was localized in the invading inflammatory cells, suggesting that VEGF-C are mainly produced by inflammatory cells in the corneal stroma, bound to VEGFR-3 on the lymphatic vessels in the conjunctiva and induced lymphangiogenesis in the injured cornea [42]. This same study also noted that VEGF-C was highly expressed in the normal conjunctiva where the lymphatic system was well developed, whereas VEGF-C was not detected in the normal cornea where no lymphatic vessels were present. This is consistent with the notion that VEGF-C is a survival factor for lymph endothelial cells. In addition, using a murine model, Narimatsu et al. also illustrated that the expression of the pro-lymphangiogenic factors VEGF-C and VEGFR-3 increased in the late stage of bacterial keratitis, which corresponded to the delayed lymphangiogenesis [38].

#### 3.4.7. Anti-VEGF Therapies in External Eye Diseases

Anti-VEGF therapies have been used in various ocular conditions, including mostly retinopathy-related conditions, and their uses have been expanded to include ocular inflammation conditions and DED [47]. Bevacizumab (Avastin; Genentech, South San Francisco, CA, USA) is a humanized monoclonal antibody used to treat angiogenesis by binding to and inhibiting the biologic activity of all kinds of VEGF subtypes in in vitro and in vivo assay systems [48]. Jiang et al. concluded that the subconjunctival injection of bevacizumab is a safe and efficient treatment for ocular surface inflammation [49]. A recent double-masked, randomized trial also illustrated significant improvement in tear-film stability, corneal staining, and symptoms in DED patients treated with bevacizumab eye drops [50]. Bevacizumab has also been recently used in meibomian gland dysfunction-associated posterior blepharitis [51], demonstrating that intra-meibomian gland injection and eye drops can significantly improve DED symptoms. Ongoing studies regarding bevacizumab on experimental CNV models have illustrated a reduction in CNV area and the number of blood vessels [52,53,54], making these existing anti-VEGF therapies a viable option for the expansion of use in external eye disease.

VEGF-C and VEGFR-3 are pathophysiologically relevant factors in corneal lymphangiogenesis and therefore are hypothetical targets for anti-VEGF C drugs. Local drug delivery and dual-functional nano-eye drops also seem to be promising developments that can be used as adjunct therapeutic options [55]. Combining the two, Lee et al. [40] examined the clinical effect of VEGFR inhibition using a murine model for AED with the administration of ovalbumin eye drops, which demonstrated a probable target for reducing clinical disease. Goyal et al. also demonstrated that treatment with anti-VEGF-C led to significant improvement in dry eye disease, indicating that targeting pro-lymphangiogenic factors can be a potential therapeutic direction for the treatment of dry eye disease [54]. There have been suggestions by Xue et al. [39] for a potential target of MIP-2 (IL-8) as a therapeutic control of *P. aeruginosa* keratitis by controlling the levels of ocular angiogenesis.

Ongoing research studies on miRNA have revealed potential therapeutic targets for VEGF in external eye disease [56]. A review by Zhang et al. summarized miRNAs into three categories based on their effects on ocular neovascularization [57]. Another review by Liu et al. also highlighted miRNAs and their potential to be targeted for treating neovascular ocular diseases [58]. Using miR-132 antagomirs, a study by Mulik et al. demonstrated that miR-132 activity regulates the lesions caused by HSV infection, suggesting an effective approach to control cornea neovascularization [59].

#### 3.4.8. Study Limitations

Individually, the studies have their own flaws. To illustrate, Philip et al. [31] were limited by the variety of etiological causes of limbal cell deficiency that were responsible for corneal neovascularization (including rejected corneal allografts, herpetic stromal keratitis, chemical burns, and atopic keratoconjunctivitis). The authors were also unable to perform age-matched comparisons. Zakaria et al. [32] did not evaluate the clinical tests for DED, and their results could potentially be affected by the confounding effect of coexisting DED in these eyes. Similarly, the crude results by Wuest et al. [35] were not adjusted for other confounding variables; particularly, some groups were significantly different in age compared to healthy controls (*p* < 0.001). Van Acker et al. [30] also had mixed overall findings regarding VEGF levels over time.

As a whole, there were no randomized controlled trials or population-based studies available. Additionally, the available studies had small sample sizes. Secondly, there could have been some selection bias introduced in this review, particularly sampling bias, which may threaten the external validity of this review and influence the generalizability of the results. the framing effect or confirmation bias can also be present. Thirdly, the use of ROBINS-E as a measure of risk-of-bias assessment is not without its flaws. Several critics have expressed practical concerns about its usage and considered its foundations to be appropriate for ‘ideal’ randomized controlled trials. A review by Bero et al. suggests that the ROBINS-E tool is too complex and does not meet the requirement for an international standard for evaluating human observational studies that investigate the risks posed to public and environmental health and proposes a simpler tool, based on empirical evidence of bias [60]. The authors do concur with the statement after personal experience of using the tool; however, other tools such as the Newcastle–Ottawa Scale (NOS) have their own flaws, including the overestimation of the quality of included studies.

The issue of potential confounding should also be mentioned. Patients with a specific ocular surface disease like pterygium or allergy may also concurrently suffer from idiopathic dry eye disease, especially in older adults. This is of particular concern if the VEGF is detected in human tears but has not been evaluated within the diseased tissue, such as the pterygium itself. One way to account for this would be to investigate clinical symptoms and signs that could help to elicit DED in addition to the presence of the evaluated condition. Participants may then be stratified into groups based on the severity or presence of DED, or the analyses statistically adjusted for these factors as covariates. The symptoms of DED and OSD, though not specific, can be characterized using standardized questionnaires and surveys such as the Ocular Surface Disease Index (OSDI) and Symptom Assessment in Dry Eye (SANDE), but not many of the reviewed studies included these measures.

Despite these limitations, the findings from this review have an impact on VEGF research. Anti-VEGF therapies have shown remarkable outcomes in reducing abnormal blood vessel growth, improving corneal clarity, and preserving visual function. This project may have public health implications and can help to assess the burden of external eye disease.

## 4. Conclusions

By summarizing the current evidence of VEGFs in external eye diseases, this review significantly increases our understanding of VEGFs and their pathological role and demonstrates a role for VEGFs in corneal neovascularization in external eye disease. More studies need to be conducted on the role of VEGFs in pseudoexfoliation syndrome. This review also highlights the efficacy of anti-VEGF therapies in managing corneal neovascularization. However, given the limitations of current research, more studies are needed to evaluate VEGFs, especially in more severe stages of OSDs without neovascularization. This is necessary for the validation of tear VEGF as a biomarker. Further research is still needed to fully understand the long-term safety and efficacy of anti-VEGF treatments.

## Figures and Tables

**Figure 1 ijms-25-01369-f001:**
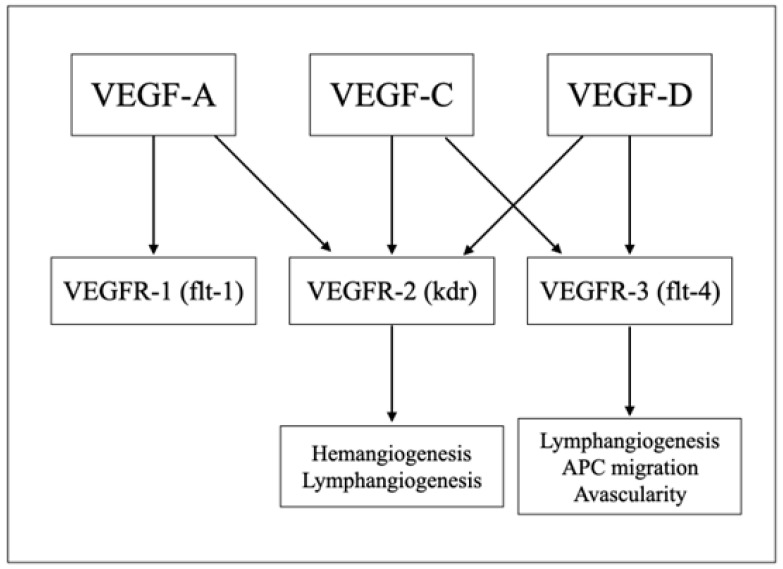
Schematic illustration of VEGFs and their receptor systems.

**Figure 2 ijms-25-01369-f002:**
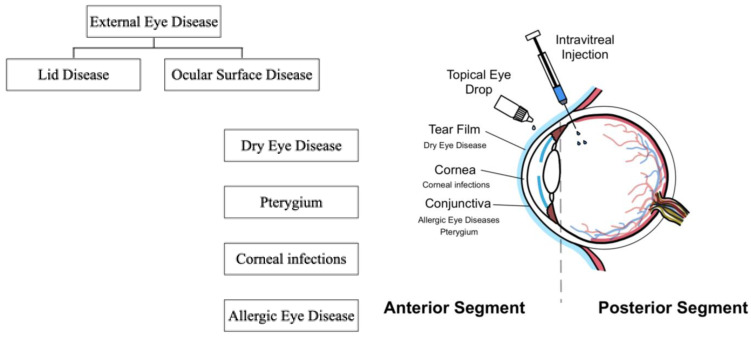
Ocular surface diseases and VEGF treatment.

**Figure 3 ijms-25-01369-f003:**
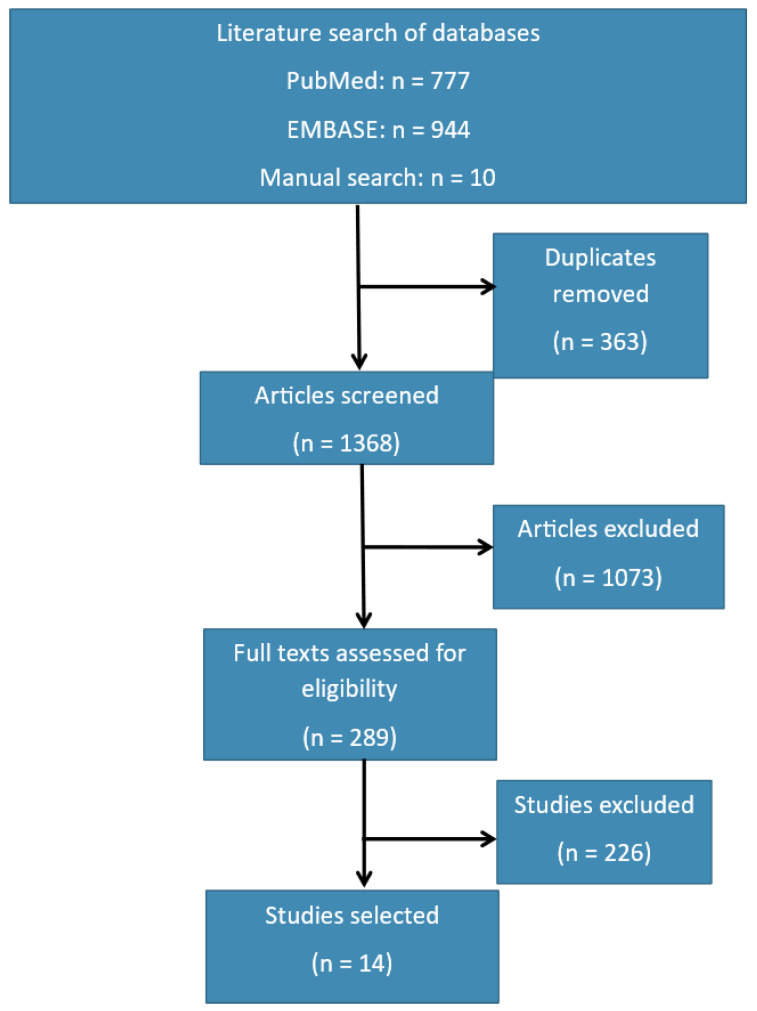
Methodology of literature search.

**Figure 4 ijms-25-01369-f004:**
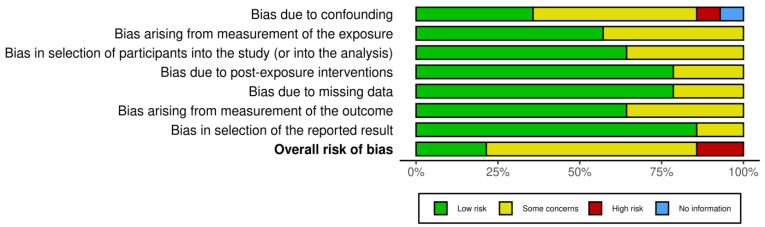
Weighted bar plots of the distribution of risk-of-bias assessments within each bias domain using the ROBINS-E tool.

**Figure 5 ijms-25-01369-f005:**
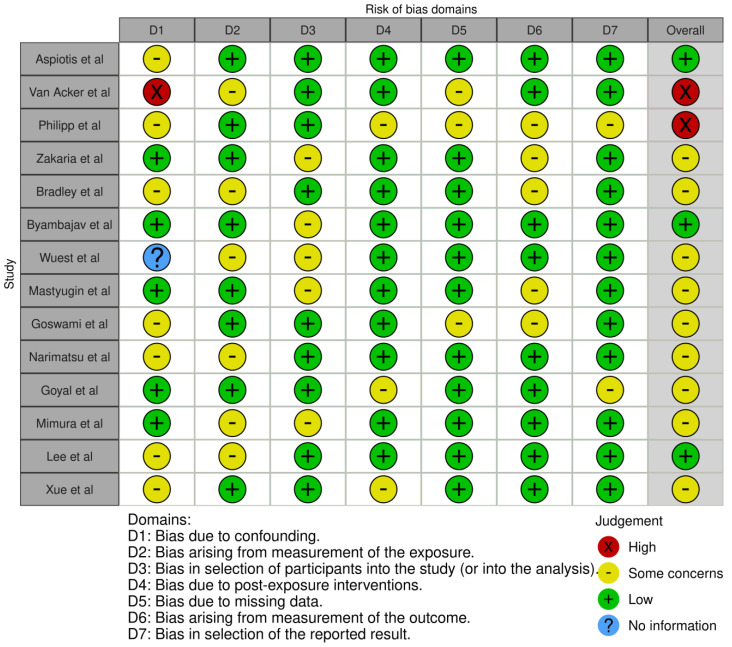
“Traffic light” plots of the domain-level assessments for each individual result [29,30,31,32,33,34,35,36,37,38,39,40,41,42] using the ROBINS-E tool [43].

**Table 1 ijms-25-01369-t001:** Summary of included studies and their respective findings.

Author, Ref., Year	Title	Model	External Eye Disease	Findings on VEGF in External Eye Disease
Aspiotis et al. [29], 2007	Angiogenesis in pterygium: study of microvessel density, vascular endothelial growth factor, and thrombospondin-1	Human-derived samples: pterygia and normal conjunctivae	Pterygium	VEGF is overexpressed in pterygium endothelial and stromal cells but not in epithelial cells, compared to normal conjunctival tissue.
Van Acker et al. [30], 2019	Pterygium Pathology: A Prospective Case–Control Study on Tear Film Cytokine Levels.	Human-derived samples: tears, pterygia	Pterygium	Proposed biomarkers did not reveal a linear relationship with corneal neovascularization nor the invasive behavior of pterygium, and no exact role in pterygium pathology could be established.
Philipp et al. [31], 2000	Expression of vascular endothelial growth factor and its receptors in inflamed and vascularized human corneas	Human-derived samples: frozen sections of cornea	Corneal neovascularization	VEGF was expressed by epithelial cells, corneal endothelial cells, the vascular endothelial cells of limbal vessels, and newly formed vessels in the stroma, and weakly by keratocytes. VEGF expression was often markedly increased in inflamed corneas. VEGF concentrations were significantly higher in vascularized corneas compared with normal control corneas (*p* < 0.001).
Zakaria et al. [32], 2012	Human tears reveal insights into corneal neovascularization	Human-derived samples: tears, cornea	Corneal neovascularization	The concentration of pro-angiogenic cytokines in human tears was significantly higher than their concentrations in serum, with the highest levels found in basal tears. Significantly higher concentrations of IL-6, IL-8, and VEGF were observed in localized corneal tears of patients with neovascularized corneas when compared to the control group.
Bradley et al. [33], 2008	Serum growth factor analysis in dry eye syndrome	Human-derived samples: serum	Dry eye disease	No statistically significant differences were observed in the serum levels of tested growth factors in DES patients versus age-matched controls.
Byambajav et al. [34], 2023	Tear Fluid Biomarkers and Quality of Life in People with Type 2 Diabetes and Dry Eye Disease	Human-derived samples: tears	Dry eye disease	Concentrations of IL-6 and IL-8 were increased in T2D-related DED. No significant difference was observed in VEGF concentrations among the 4 groups (*p* = 0.5).
Wuest et al. [35], 2011	The herpes simplex virus-1 transactivator-infected cell protein-4 drives VEGF-A dependent neovascularization.	HSV-1infection in Tert-immortalized human corneal epithelial cells	Corneal neovascularization induced by HSV1	HSV-1-infected cells were the dominant source of VEGF-A during acute infection.
Mastyugin et al. [36], 2001	Corneal epithelial VEGF and cytochrome P450 4B1 expression in a rabbit model of closed eye contact lens wear	Rabbit model of closed-eye contact lens-induced injury	Hypoxic injury	VEGF mRNA levels were elevated in the hypoxic corneal epithelium.
Goswami et al. [37], 2021	Pathophysiology and inflammatory biomarkers of sulfur mustard-induced corneal injury in rabbits.	Rabbit model	Sulfur mustard-induced corneal injury	VEGF expression was increased and remained significantly elevated till day 14 post-exposure. ~6.0- and 4.6-fold increases, respectively, were observed for 5 min and 7 min SM exposure groups compared to the control group.
Narimatsu et al. [38], 2019	Corneal lymphangiogenesis ameliorates corneal inflammation and edema in late stage of bacterial keratitis	Mouse bacterial keratitis model using Pseudomonas aeruginosa	Bacterial keratitis Corneal lymphangiogenesis	Corresponding to delayed lymphangiogenesis, the expression of the pro-lymphangiogenic factors VEGF-C and VEGFR-3 increased in the late stage of bacterial keratitis.
Xue et al. [39], 2002	Macrophage inflammatory protein-2 and vascular endothelial growth factor regulate corneal neovascularization induced by infection with Pseudomonas aeruginosa in mice.	Mouse bacterial keratitis model using Pseudomonas aeruginosa	Corneal neovascularization induced by infection with Pseudomonas aeruginosa	Expression of MIP-2 and VEGF was significantly (*p* < 0.05) elevated after bacterial infection.
Lee et al. [40], 2015	Involvement of corneal lymphangiogenesis in a mouse model of allergic eye disease.	Mouse model of allergic eye disease	Allergic eye disease	Corneas in AED showed a significant upregulation of the lymphangiogenic ligands VEGF-C and VEGF-D, as well as a nearly seven-fold increase in mRNA levels of VEGFR-3.
Goyal et al. [41], 2010	Evidence of corneal lymphangiogenesis in dry eye disease: a potential link to adaptive immunity?	Mouse model	Dry eye disease Corneal lymphangiogenesis	Lymphangiogenic-specific VEGF-D and VEGFR-3 levels were increased, earliest on day 6, followed by increased VEGF-C, VEGF-A, and VEGFR-2 levels. Low-grade inflammation associated with DED was found to be an inducer of lymphangiogenesis without accompanying hemangiogenesis.
Mimura et al. [42], 2001	Expression of vascular endothelial growth factor C and vascular endothelial growth factor receptor 3 in corneal lymphangiogenesis	Mouse model	Corneal lymphangiogenesis	Expression of VEGF-C mRNA in the rat cornea was dramatically upregulated 3 days after the injury.VEGFR-3 expression in the rat cornea was minimally detected before the injury and was upregulated 3 and 7 days after the injury. VEGF-C and VEGFR-3 were found to be pathophysiologically relevant endogenous factors in corneal lymphangiogenesis.

**Table 2 ijms-25-01369-t002:** Overview of each individual study’s characteristics.

Author, Ref., Year	Study Methods	Participants	Intervention
Aspiotis et al. [29], 2007	Retrospective Case–control Single center Human-derived samples	n = 52 (Male = 22, Female = 30) Patients from a single hospital Age: 59–82 years-old	Fifty-two surgically excised pterygia and seven normal conjunctivae were immunohistochemically studied.
Van Acker et al. [30], 2019	Prospective Age/gender-matched Human-derived samples	Total n = 38: 19 patients (Male = 11, Female = 8); 19 controls. Age range: 23–78 years-old	This study enrolled 19 patients and age/gender-matched healthy controls. Tear-film levels of interleukin (IL)-6, IL-8, and vascular endothelial growth factor (VEGF) were investigated over time, and preoperative concentrations were linked to corneal neovascularization and pterygium size.
Philipp et al. [31], 2000	Case–control Human cornea	Total sample obtained: 38; however, n = 13 included: Corneas with significant vascularization in at least 2 or more quadrants	Polyclonal antibodies to VEGF and its receptors were used for the immunohistochemical staining of frozen sections of 38 human corneas with various degrees of neovascularization and inflammation.
Zakaria et al. [32], 2012	Observational Case–control Human cornea/tear samples	12 patients with corneal neovascularization; 10 healthy volunteers	Basal tears along with reflex tears from the inferior fornix and superior fornix, using a corneal bath, were collected along with blood serum samples. Concentrations of the pro-angiogenic cytokines interleukin (IL)-6, IL-8, VEGF, monocyte chemoattractant protein 1 (MCP-1), and Fas ligand (FasL) were determined in blood and tear samples using a flow cytometric multiplex assay.
Bradley et al. [33], 2008	Age/Gender-matched Human-derived samples	Total n = 12 (6 patients, 6 controls) All female	Six female dry eye syndrome patients and six age- and gender-matched controls were included. Enzyme-linked immunosorbent assays were performed to quantify serum growth factor levels.
Byambajav et al. [34], 2023	Case–control Human-derived samples	Total n = 122: Healthy control n = 17; DED-only (without T2D) n = 17; T2D-only (without DED) n = 41; DED + T2D n = 47	Unstimulated basal tears were collected from each subject. Metabolic proteins, inflammatory cytokines concentrations, and final protein concentrations were analyzed and measured.
Wuest et al. [35], 2011	Case–control Murine model	Total n = 12 n = 3/group/experiment	Either C57BL/6 controls or mice with a floxed VEGF-A gene (Flxd-VEGFA) were infected with either wild-type HSV-1 strain SC16 or a Cre-expressing HSV-1 recombinant virus, which was derived from strain SC16 and expressed Cre under the control of the HSV-1 ICP0 promoter.
Mastyugin et al. [36], 2001	Case–control Rabbit model	n = 4 Right eye (OD) was utilized leaving the left eye (OS) for control	Rabbit eyes were fitted with contact lenses followed by a silk suture tarsorrhaphy. The anterior surface was analyzed at 2, 4, and 7 days via slit lamp biomicroscopy, subjective inflammatory scoring, and corneal pachymetry. Corneal epithelium was scraped, and CYP4B1 and VEGF mRNA levels were measured.
Goswami et al. [37], 2021	Case–control Rabbit model	n = 6–7/per treatment group at each time point	The left eye of the animal was treated as control (naïve or untreated), and the right eye was exposed to 400 μg/L SM (~390–420 μg/L; n = 6–7/per treatment group at each time point) for 5 min (short-duration exposure) or 7 min (long-duration exposure).
Narimatsu et al. [38], 2019	Case–control Murine model	n = 30 (n = 10/group)	Pseudomonas aeruginosa strain was inoculated onto the mouse cornea. Control mice were inoculated with phosphate-buffered saline (PBS). These corneas were harvested from these mice on days 2, 7, and 14 post-inoculation. Corneal angiogenesis and lymphangiogenesis were evaluated via immunostaining and examined using fluorescence microscopy.
Xue et al. [39], 2002	Case–control Murine model	Total n = 12: Control mice (n = 6); Corneas of infected mice (n = 6)	After administering an anti-MIP-2 antibody or control antibody, mouse corneas were challenged with *P. aeruginosa*. The expression of MIP-2 and VEGF was detected using an ELISA from ocular homogenates.
Lee et al. [40], 2015	Case–control Murine model	Whole-mounted corneas (n = 4–7/group) Excised corneas (n = 3/group)	Allergic eye disease was induced by ovalbumin (OVA) immunization and chronic OVA exposure. Confocal microscopy of LYVE-1-stained cornea allowed for an evaluation of corneal LA, and qRT-PCR was used to evaluate the expression of VEGF-C, -D, and -R3 in these mice. Administration of VEGF receptor (R) inhibitor was incorporated to inhibit corneal LA in AED. Immune responses were evaluated by in vitro OVA recall responses of T cells and IgE levels in the serum.
Goyal et al. [41], 2010	Case–control Murine model	Total n = 20 Five mice (10 eyes) were included in each group. Two corneas were pooled together to equal 1 sample	Dry eye was induced in murine eyes using high-flow desiccated air. Corneas were double-stained with CD31 (panendothelial marker) and LYVE-1 (lymphatic endothelial marker).
Mimura et al. [42], 2001	Case–control Murine model	Total n = 75 For each time point, three eyes were used for electron microscopy, three for immunohistochemistry, and nine for competitive reverse transcription polymerase chain reaction (RT-PCR) Timepoints: Before treatment and at 3 days, 1 week, 2 weeks, and 4 weeks after injury	A rat model was used in which silver nitrate application resulted in corneal circumferential neovascularization. Electron microscopy, reverse transcription polymerase chain reaction, antibodies, and immunohistochemistry were used.

## Data Availability

No new data were created or analyzed in this study. Data sharing is not applicable to this article.

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
