# Peer review of "A Systematic Review of Tear Vascular Endothelial Growth Factor and External Eye Diseases"

_ijms, 2024, doi:10.3390/ijms25031369_

Round 1
Reviewer 1 Report
Comments and Suggestions for Authors
Although the review of tear VEGF might be original, there are a lot of flaws which have to be resolved to improve the manuscript.
I wonder why the author mix the VEGF tear analysis in ocular surface disease (pterygium, Dry Eye disease, corneal infections, Allergic Eye Disease) and PEX glaucoma. If the review focus only VEGF tear in ocular surface disease, it would be more interesting for readers. The structure and organisation of the manuscript should be improved in a logical sequence.
1. Introduction : The physiopathology of pterygium, Dry eye disease and allergic eye diseases is lacking. State the origin of VEGF tear (expression of VEGF by corneal epithelial cells, which factors increase its secretion..)
2. Methods: explain the choice of exclusion of interventional studies
3. Results: The table is poorly presented. Findings are not readable.
Table 1: I would suggest separating human studies/ animal studies/ Type of eye diseases
Table 2 : The table is also poorly presented. Author, year ref should be in the same column. The last column is not readable. Only short results need to be shown. Not sure that the country is useful to be cited.
Organisation of the section 3.2 needs to be improved
4. Discussion: needs to be organized as follow
- General findings in tear VEGF in human: age, sex, relation between tear VEGF and serum VEGF
- Findings from studies : in Pterygium, in corneal vascularization, in DED, in corneal infection. Distinguish corneal lymphangiogenesis in acute/ sequelae phase
- summarize results for each disease
Author Response
Please refer to this attachment

Reviewer 2 Report
Comments and Suggestions for Authors
The presented review article allows us to evaluate the results and effectiveness of anti-VEGF therapy in the treatment of corneal neovascularization.
The volume of work carried out by the author is impressive - the initial literature search yielded 777 PubMed, 944 Embase and 10 studies from manual search. If necessary, the authors of the relevant articles were contacted. Summary of included studies and their respective findings and overview of each individual study characteristics conveniently presented in table format.
Of particular importance is that the authors assessed the quality of the selected studies, including an assessment of the risk of bias and risk of systematic error. Another advantage of this review is that the authors describe in detail the chapter “4.2 Study limitations”.
This work does not contain fundamental comments, the topic under study is novel and can be recommended for publication in open sources.
Author Response
Please refer to this attachment

Round 2
Reviewer 1 Report
Comments and Suggestions for Authors
The manuscript is clear and readable now. Thank you for your efforts